# Implementing in-situ self-organizing maps with memristor crossbar arrays for data mining and optimization

Rui Wang[1,2,3], Tuo Shi[1,3,4✉], Xumeng Zhang [2], Jinsong Wei[1,4], Jian Lu[1,4], Jiaxue Zhu[1,3], Zuheng Wu[1,3], Qi Liu [1,2,3✉] & Ming Liu[1,2,3]

A self-organizing map (SOM) is a powerful unsupervised learning neural network for analyzing high-dimensional data in various applications. However, hardware implementation of SOM is challenging because of the complexity in calculating the similarities and determining neighborhoods. We experimentally demonstrated a memristor-based SOM based on Ta/TaO$_x$/Pt 1T1R chips for the first time, which has advantages in computing speed, throughput, and energy efficiency compared with the CMOS digital counterpart, by utilizing the topological structure of the array and physical laws for computing without complicated circuits. We employed additional rows in the crossbar arrays and identified the best matching units by directly calculating the similarities between the input vectors and the weight matrix in the hardware. Using the memristor-based SOM, we demonstrated data clustering, image processing and solved the traveling salesman problem with much-improved energy efficiency and computing throughput. The physical implementation of SOM in memristor crossbar arrays extends the capability of memristor-based neuromorphic computing systems in machine learning and artificial intelligence.

[1] The Key Laboratory of Microelectronics Devices and Integrated Technology, Institute of Microelectronics Chinese Academy of Sciences, 100029 Beijing, PR China. [2] The Frontier institute of Chip and System, Fudan University, 200433 Shanghai, PR China. [3] University of Chinese Academy of Sciences, 100049 Beijing, PR China. [4] Institute of Intelligent Computing, Zhejiang Laboratory, 311122 Hangzhou, PR China. ✉email: shituo@ime.ac.cn; qi_liu@fudan.edu.cn

Neuromorphic computing systems built with memristors could have much-improved power efficiency and computing throughput than traditional hardware[1–6]. Recent implementations with memristors are, however, primarily artificial neural networks for supervised learning algorithms[7–13]. Unsupervised learning networks inspired by biological systems can learn through data sets without labels and hence are more energy- and cost-efficient[14–16]. Compared with the supervised network, the unsupervised learning are more similar with human brain and has more extensive application, considering that most data and information are unlabeled in real world. Besides, unsupervised approach can cluster or pre-process the unlabeled complex data to smaller subspaces for subsequent classification through another supervised network. A self-organizing map (SOM), also called a 'Kohonen network', is a frequently used unsupervised algorithm inspired by the topological maps in the sensory-processing areas of the brain, where neurons responding to similar inputs are spatially located very close[17,18]. As a result, SOMs can identify relationships of input data and are well suited for clustering and optimization problems such as language recognition and text mining, financial predictions, and medical diagnosis[19–24]. Furthermore, SOM is a nonlinear dimension-reduction tool that automatically maps high-dimensional data to a lower dimension (usually two- or one-dimensional), more effectively in nonlinear distributions than classical linear algorithms such as multi-dimensional scaling[25] or principal components analysis[26].

However, implementing SOM in conventional CMOS-based hardware is limited by the complexity in calculating the similarities and determining neighborhoods, which imposes an enormous increase in computing time and power consumption as the number of neurons and features increase[27]. It is therefore imperative to seek emerging energy-efficient hardware with parallel computing capacity for SOM networks. Memristor, a two-terminal resistance switch with multiple conductance states as synaptic weights, has been organized into large-scale crossbar arrays to implement parallel and energy-efficient in-memory computing using physical laws[28–32]. On the other hand, experimental demonstrations of SOM using memristors are yet to be achieved due to two main challenges: finding the shortest Euclidean distance with the unnormalized vectors and implementing complex topology of SOM output layer without extra hardware cost[33,34].

Herein we report our experimental implementation of SOMs in a $128 \times 64$ 1-transistor 1-memristor (1T1R) crossbar array and its applications in data mining and optimization. The similarities between inputs and weight are directly calculated through Euclidean distance in the hardware. The neighborhood function of SOM is directly realized by the topological structure of the memristor array without extra circuits. Memristor-based 1D-SOM and 2D-SOM are successfully employed to solve color clustering and traveling salesman problems. Compared with traditional hardware, the memristor-based SOM system has better power efficiency and higher parallelism in computing, extending the application range of memristor-based neuromorphic computing systems in artificial intelligence.

## Results
### SOM topography and algorithm
An SOM is composed of an input layer and an output layer (Fig. 1a). The input layer has multiple dimensionless nodes that act as input neurons, while the output layer is a one-dimensional (1D) line or two-dimensional (2D) grid of neurons for 1D and 2D SOM, respectively (Fig. S1). Each input node is connected to every output neuron through a weighted connection (synapses). Unlike an ordinary fully-connected unsupervised neural network, the output neurons in SOM can communicate to their topological neighborhoods. For example, for a given pattern, the best matching unit (BMU) is determined by finding the neuron with the synaptic weight vector that is most similar to the input vector. During weight updating, both the synapse connected to the BMU and those connected to the BMU's topological neighborhoods are modified to increase the strength of the match. The weights of neighborhoods are determined by the distance between the BMU and neighborhood neurons through the neighborhood function. Typically, the neighborhood function is a unimodal function that is symmetric around the winner's location. The connection strength decreases monotonically with the distance from the winner. Response of neurons to similar inputs is spatially located very close, among which typical ones are shown in Supplementary Fig. S1. Due to the unique neighborhood function of the SOMs, this mapping retains the relationship between input data as faithfully as possible, thus describing a topology-preserving representation of input similarities in terms of distances in the output space[35].

Euclidean distance, which refers to the distance between the input vector (feature vectors) and weight matrix (dictionary vectors) in a Euclid space, represents similarities between two vectors. A smaller Euclidean distance between two elements means they are more similar, and the smallest Euclidean distance corresponds to the BMU[36,37]. For a certain n-dimension input $X$ and an $n \times m$ weight matrix $W$, the Euclidean distance is calculated by

$$D = ||X - W||^2 = X^2 - 2W \cdot X + W^2 \tag{1}$$

After finding the BMU, the weights of the winner and its neighborhoods are updated based on the following equations.

$$\Delta W = \eta \cdot T_i \cdot (X - W) \tag{2}$$

Where $\eta$ is the learning rate of matrix $W$ and $T_i$ is the neighborhood function of the $i$th neuron determined by

$$T_i = \exp\left(-\frac{(r_c - r_i)^2}{2 \cdot \delta}\right) \tag{3}$$

In this equation, $r_i$ and $r_c$ represent the location of neuron $i$ and winner $c$, $(r_c - r_i)^2$ denotes the topological distance between neuron $i$ and the winner neuron, and $\delta$ is a time-carrying parameter that guides the reduction of the neighborhood function during training.

### Implementing SOM in a memristive crossbar array
We experimentally implemented the SOMs in a $128 \times 64$ 1T1R memristor array composed of Pd/TaO$_x$/Ta memristors (Fig. 1b, c). The programming and computing were implemented by off-chip peripheral circuits on custom-built print circuit boards (PCBs) and MATLAB scripts, as have been successfully used in our early demonstrations[38–40]. The 1T1R memristor array acted as a weight matrix ($W$), consisting of two types of weights ($W_{data}$ in data rows and $W_{squared}$ in square rows). In our SOM systems, all the weights are linearly mapped into the conductance of the memristors, as shown in Fig. 1d. In the computing, normalized input vectors (0–1) to the data rows were coded with amplitudes of voltage pulses of fixed pulse width.

The BMU is detected by finding the shortest Euclidean distance between the $W_{data}$ and input vectors. Only if the input vectors and $W_{data}$ are both normalized, the shortest Euclidean distance is equivalent to the smallest dot product between the input vectors and $W_{data}$. Unfortunately, numerically normalizing the weight vectors after every updating step is difficult and resource-demanding. In our SOM, we use the square row method to calculate the Euclidean distance without normalizing the weight matrix in every step. For a certain input vector $X$, $X^2$ was a constant;

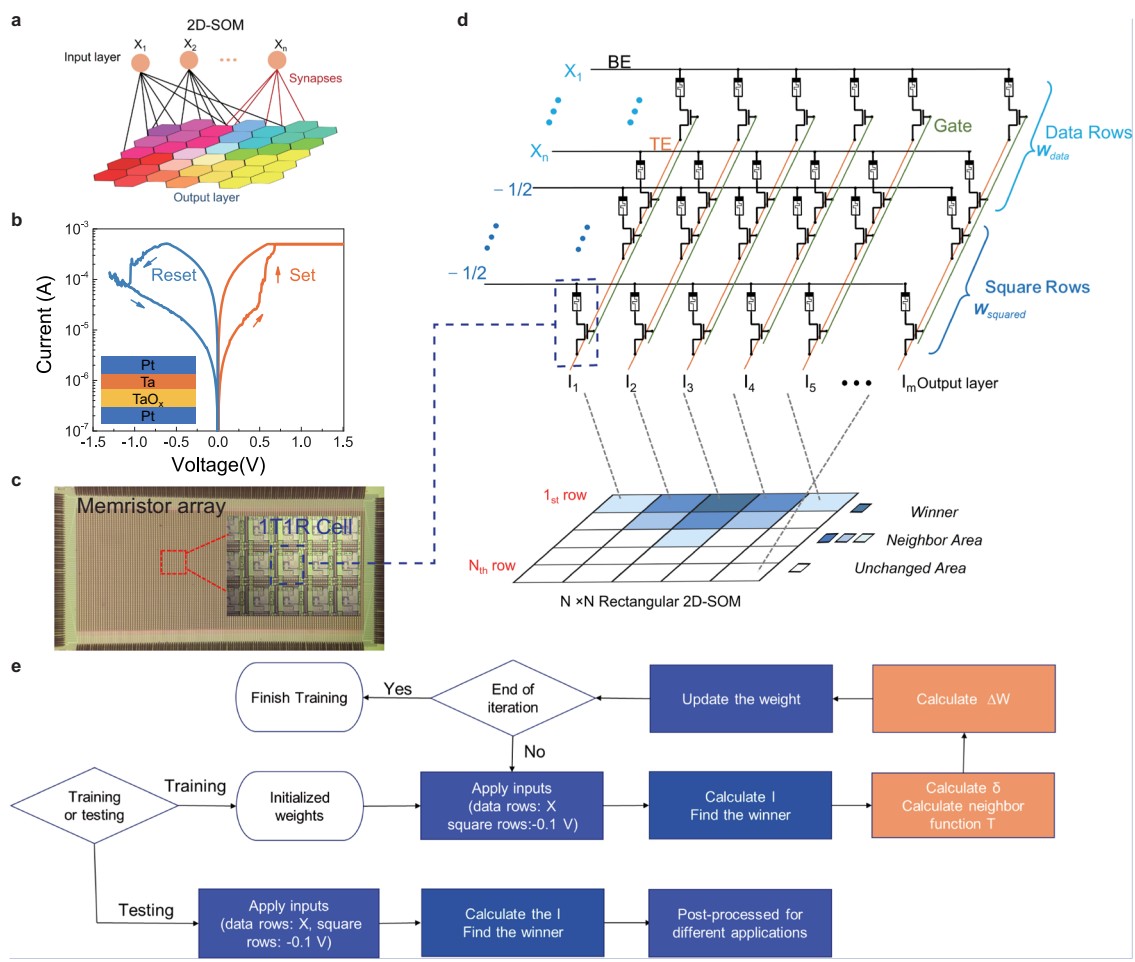

**Fig. 1 The SOM concept and its implementation with memristor crossbar arrays. a** Schematic of a 2D SOM that is composed of a unidimensional input layer and a bidimensional output layer. Each node in the input layer is connected to all nodes in the output layer through the synapses. The output nodes can communicate with their neighborhoods to form a 2D-planer topological map. **b** Typical I–V characteristic of a memristive device used as synapses in the SOM. The device size is 2 µm by 2 µm, and the material stack is shown in the inset. **c** Image of a 128 × 64 1T1R memristor array (Scale bar, 500 µm). Inset shows a close-up image for part of the chip marked by a red box, and the blue dashed box shows one 1T1R cell (scale bar, 10 µm). **d** The implementation of the 2D SOM with the 1T1R memristor array. The synaptic weights are stored as the memristor conductance in data rows. The squared values are stored in the square rows. The top electrode (TE), bottom electrode (BE), and gate line are connected to the off-chip peripheral circuits for precise weight updating. The Euclidean distance between the input vector and weight vector is calculated in one reading process. The output currents in different columns represent the signal applied to different output nodes. The intrinsic topology of the nodes in the memristor array is a 1D line. And 2D topology of the SOM is artificial defined. For example, the output nodes of 1st to the 5th columns in the memristor array act as the nodes in the first row of 2D 5 × 5 output layer, and the last five columns in the memristor array act as the last row of the 2D 5 × 5 SOM. The node with the largest current is the winner, whereas the nodes close to the winner (in 2D space) are the neighborhood. **e** Flow chart of the training and testing processes. Steps in blue boxes are implemented in hardware in this work, while those in orange boxes are achieved in a CPU.

$W_{data} \cdot X$ term was calculated using the standard vector-matrix multiplication approach through Ohm's Law and Kirchhoff's Current Law in a crossbar array. During the training process, in each training epoch, for all nodes the $X^2$ term are only depended on the input value and equal with each other, as a result the $X^2$ term will not affect the comparison when determining the winner. However, the **W** will be updated in each epoch, for different node the weight is different. As a result, the $\mathbf{W}^2$ term affects the selection of the winning neuron and cannot be neglected. To calculate the $\mathbf{W}^2_{data}$ term, the inputs to the square rows were kept at $-1/2$ (represented by $-0.1$ V as the read voltage was 0.2 V), and the weights of the square rows in a certain column (for example,

$m^{th}$ column in the array) were all the same ($w_{m\_square} = \frac{\sum_{i=1}^{n} w^2_{data\_im}}{l}$, where $l$ is the number of square rows and $n$ is the number of data

rows). As a result, the output current at the $m^{th}$ column is $I_m = W_{data} \cdot X - l \cdot \frac{W_{squared}}{2} = \frac{1}{2}(2 \cdot W_{data} \cdot X - W^2_{data})$, opposite to the Euclidean distance between the input vector and the weight. As a result, the column with the largest measured current has the smallest Euclidean distance with the input, and the corresponding neuron will be the winner or the BMU. We used multiple additional rows in the crossbar (square rows in Fig. 1d), ensuring that the memristors in the data rows and square rows have the same conductance dynamic range. In the most extreme situation (all the weights in a column are the maximum value "1"), the largest number of the additional square rows $l$ should be equal to the number of data rows $n$; so the largest weight in $W_{squared}$ will not go beyond the largest weight in $W_{data}$, while in most cases, $l$ is smaller than n. In previous one-row method, which use width (0~T) to code the input (0~1)[41], to fit the conductance dynamic range of data row and

additional row, the conductance of the device in the one additional row is $\frac{W^2}{l}$. As a result, the pulse width of the additional row should be $l \times T$, which expands $l$ times time cost than our multiple rows method. Besides, we use multiple devices to represent the $W^2$, which can significantly reduce the impact of the writing errors of one device. Compare with one-row method, our multiple square row method needs a larger area but has less time cost and less disturbance from writing errors.

After finding the BMU, the weight updating depends on the topological distance of the neurons in the output map and the neighborhood function. The intrinsic topology of the memristor-array-based neurons is 1D-line because devices in the same column act as the synapse connecting to one neuron in the array. Figure 1d presents the schematic of utilizing this 1D-topology to form a 2D rectangular SOM output map. For a $5 \times 5$ SOM, the first five columns of the array act as the first row of the output map; the 6th to 10th columns of the array act as the second row of output map; and the last five columns act as the last row of the output map. The weights in the data rows ($W_{data}$) will be updated according to Eq. (2), with a gaussian neighborhood function without the depression part has been used in our SOM, and those ($W_{squared}$) in the square rows will be updated by

$$\triangle W_{squared} = \frac{W_{data}^2}{l} - W_{squared} \qquad (4)$$

Because of the limitation of our hardware-measure system that can only read and write in one direction (Write on rows and read on columns), the square rows are calculated in software part in offline mode. The squared term can be directly calculated in on-line mode based on the memristor array[41]. The squared values can be obtained in the memristor crossbar by sequential backward and forward read operations through the $W$ matrix. First in backward reading process, a reading pulse proportional to 1 are applied to the n column. The scaled weight elements of data rows can be obtained while the square rows were floated. Then forward reading process, these values are then used as inputs that are fed into the data rows whereas floating the square rows. The output collected at column n then corresponds to the squared value. The network will be trained to achieve a self-organizing map by repeating the iterative training process with the search steps and the update steps. In the testing process, inputs from testing sets are applied in the arrays one by one, and the best matching unit with the biggest output current will be found. Different post-processing methods are adopted for various applications, such as transferring the weight to RGB pixel for image processing or sequence the neurons for solving TSP. The flow chart of training and testing processes is illustrated in Fig. 1e. It is worth pointing out that calculating the $\triangle W_{data}$, $\triangle W_{squared}$ and neighborhood function $T_i$ are computationally expensive. As a result, they are achieved in software and will be accomplished with circuits integrated onto the chip in the future.

**Color clustering and image processing**. Clustering is the analysis and organization of a collection of patterns through the similarities[42]. In our SOM, the similarities between input data are mapped into the output neurons' topographical or spatial organization relation, which gives the SOMs adjustable clustering ability. We used the memristor-based SOM for color clustering, and clustered 256 colors into an $8 \times 8$ SOM with a $5 \times 64$ crossbar array. The inputs to three data rows were the normalized R (red), G (green), B (blue) (0–1) color components of a pixel, and inputs to the two square rows were $-1/2$. A randomly selected color was first applied to the data rows during training, and the weights were updated according to the algorithms as described in Eqs. 2–4. The evolution of the weights is illustrated in Fig. S2, and

the final conductance map after the training process is shown in Fig. 2a. In the clustering process, a color vector was applied to each data row, $-1/2$ was still applied on the square rows. The 1st, 2nd, and 3rd rows are data rows, representing the red, green, and blue color components of output neurons, respectively. To implement a 2D-rectangular output map, devices in the 1st to the 8th columns of memristor array act as eight output neurons' weights in the first row of the 2D $8 \times 8$ output map, the devices in the 9th to the 16th columns of the array act as the second row of the output map, whereas the devices in the 57th to 64th columns representing the weights of output neurons in the last row of the output map. Figure 2b shows the self-organized pattern map of the trained color mapping SOMs. After training, nearby neurons tend to respond to the same input type colors, e.g., reds are concentrated to the bottom left corner, yellows are concentrated to the bottom right corner, and oranges are located in the middle bottom, just between the reds and yellows. The clustering ability of the SOMs is further proven on standard cluster databases, as shown in Figs. S3 and S4. And after the clustering process, the memristor-based 2D-SOM/1D-SOM system enables 94.6% and 95% high classification accuracy for IRIS data set and wine data set, respectively. Compared with supervised method (96.08% and 94.7% for IRIS and wine data set, respectively)[43,44], our SOM can achieve competitive accuracy, which is also better than memristor-based unsupervised K-mean system (93.3% for IRIS data set). Besides, in comparison with other unsupervised cluster algorithm (e.g., K-means) SOM does not need any additional information at all. K-means needs to determine the number of classes in prior. Therefore, SOM is less affected by initialization and has more generality. In training process, SOM updates the winner and the adjacent nodes. Most approach only updates a single node. Therefore, SOM is less affected by noise data. SOM also has good visualization and elegant topology diagram, which make people easier to understand and analyze[45].

The color mapping capability of our SOM, characterized by the number of output colors, is affected by the strength of the neighborhood function. The strength is evaluated by the neighborhood function factor $A$, which is represented as the initial size of neighborhood area. Based on the SOMs, a dynamic color mapping capability ratio (or resolution in color clustering) can be easily achieved by tuning the neighborhood function factor in the software part for different applications, as shown in Fig. 2c. More neighborhood neurons' weights will be updated with a larger neighborhood factor. With a large neighborhood function factor (from 5 to 50), the size of the cluster will be increased, and obviously, the number of the final clusters will be decreased. However, a sizeable compression ratio achieved by a smaller number of clusters may cause serious data loss and affect the following pattern recognition. It is worth noting that when the neighborhood function value is tiny (e.g. 0.1), only the weights of the winner will be updated. The total number of clusters is decreased since the SOMs lose the ability to recognize mixed colors from the neighborhood function.

Compared with other methods[46] (Fig. S5), the square row approach for calculating Euclidean distance in our SOM has a stronger color mapping ability (a higher resolution in color clustering). Figures 2d and S6 show the comparison of three similarity calculation methods, Euclidean distance, dot product, and normalized dot product, in the task of clustering 256 colors into 64 neurons with identical parametric configuration for the highest resolution. The Euclidean distance method shows the best result as 256 colors are successfully compressed into 48 colors with 48 firing neurons. In contrast, only 6 and 9 colors are left with dot product and normalized dot product. That is because the clustering potential is strongly related to the accuracy of the similarities obtained by the memristor array,

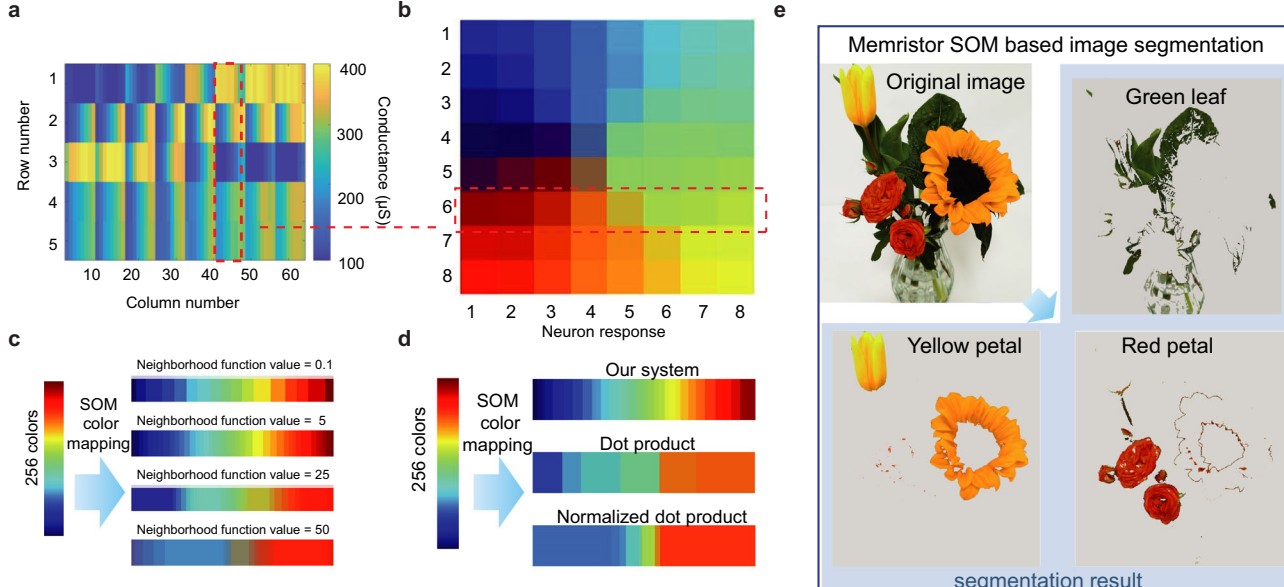

**Fig. 2 Memristor-based SOMs for color mapping. a** The conductance map of the 5 × 64 memristor array after the training process. The weights in the 1st to 3rd row are $W_{data}$, representing the R, G, B components of the output nodes, while the weights in 4th to 5th are the $W_{squared}$. **b** The self-organized topology pattern of the trained color mapping SOM. Each pattern is represented as the output neuron response. The chosen eight weight vectors in the memristor array (in the red box of **a**) represents the weights of the nodes in the 6th row of 2D SOM. The color of the output nodes in the 6th row of 2D SOM (red box in **b**) can be defined by the R, G, B components in the of weight vectors in the memristor array (red box of **a**). **c** Effect of the neighborhood function factor (from 0.1 to 50) on the mapping of 256 colors. The number of mapped colors is increased at first and then decreased with the increasing of the neighborhood function factor. **d** Comparison of our SOM, dot product, and normalized dot product methods to calculate similarities for color mapping. The number of mapped colors based on our SOM is larger than the number of other methods, which means our SOM has a stronger color mapping ability. **e** SOM-based image segmentation. Three different clusters are defined with the topology of output neurons. The original image has been segmented into three different sub-images, while the pixels in one sub-image are the nodes in the same cluster. The original image was segmented into three sub-images by different colors, while the flowers can be segmented red petal, yellow petal, and green leaf.

and our approach can directly calculate the precise Euclidean distance.

These results support that our memristor-based SOM shows good performance in different image processing tasks, including compression (Fig. S7) and segmentation. As shown in Fig. 2e, using the self-organizing feature of SOM, color-based image segmentation can be realized. After training, the neurons in the output layer are clustered into five categories according to each neuron's position and response color in the SOM (Fig S8). Pixels in the colorful origin image acts as the input and are applied into the network and classified according to the category of the corresponding response neuron. Three sub-images were obtained, while the pixels in one sub-image are the nodes in the same cluster. As a result, the original image was successfully segmented into three sub-images by different colors (red, green, and yellow). As a result, the flowers can be segmented into different parts: red petal, yellow petal, and green leaf (Fig. 2e). It is worth noting that two other parts (black stamen and white background) of the original figure can be also segmented based on the result in Fig. S8. To show the energy-efficient nature of our memristor-based SOM in image processing, we performed an energy estimation, in which the energy consumption of the array is ~3.84 μJ for the image segment tasks, which is much lower than that in CMOS platforms[27] (see Supplementary Note 1).

**Solving traveling salesman problem.** Our memristor-based SOM can be used to address optimization problems, such as the traveling salesman problem (TSP), that intends to find the shortest yet most efficient route among multiple nodes. In clustering application, people normally use the SOM with 2D planer output

layer to maintain more topology information of input data. However, in solving TSP problems, the SOM with 1D-ring output layer is more common. In clustering, the SOM is a mapping tool which maps the high dimension data to low dimension space, the neighborhood function is used to maintain the topology information of input data. In TSP, SOM is considered as an elastic ring, with the training process, the nodes turn to catch the cities, and due to the shrink ability from the neighborhood function, the ring trends to minimize the perimeter. The principle of solving TSP by memristor-based SOM is illustrated in Fig. 3a. The topology of SOMs' output layer is a 1D ring, representing the calculated route of TSP. In our memristor-based SOM, the last column of the array is considered as the neighborhood of the first column, like a set of neurons joins together in a one-dimensional ring. During the training process, the coordinates of the cities were applied to the data rows. The winner neuron, representing the closest to the chosen city, was found by detecting the column with the largest output current. The weight of the winner and its neighborhood were updated according to Eqs. 2–4. It seems that the winner node moves in the city plane and induces its neighborhood on the ring to do so, but with a decreasing intensity along the ring. This correlation between the motion of neighborhood nodes intuitively leads to a minimization of the distance between two neighborhoods, hence giving a short tour[47]. In the testing process, all cities are applied to the SOM one by one, normally different winners were found when applying different cities. As shown in Fig. 3a, for example, in a four-city TSP, the coordinate will be applied one by one, while the 2nd node is the winner when city A is applied, whereas the 1st, the nth, and the 4th nodes are the winners when city B, city C, and city D are applied, respectively. The intrinsic physical sequence of nodes can

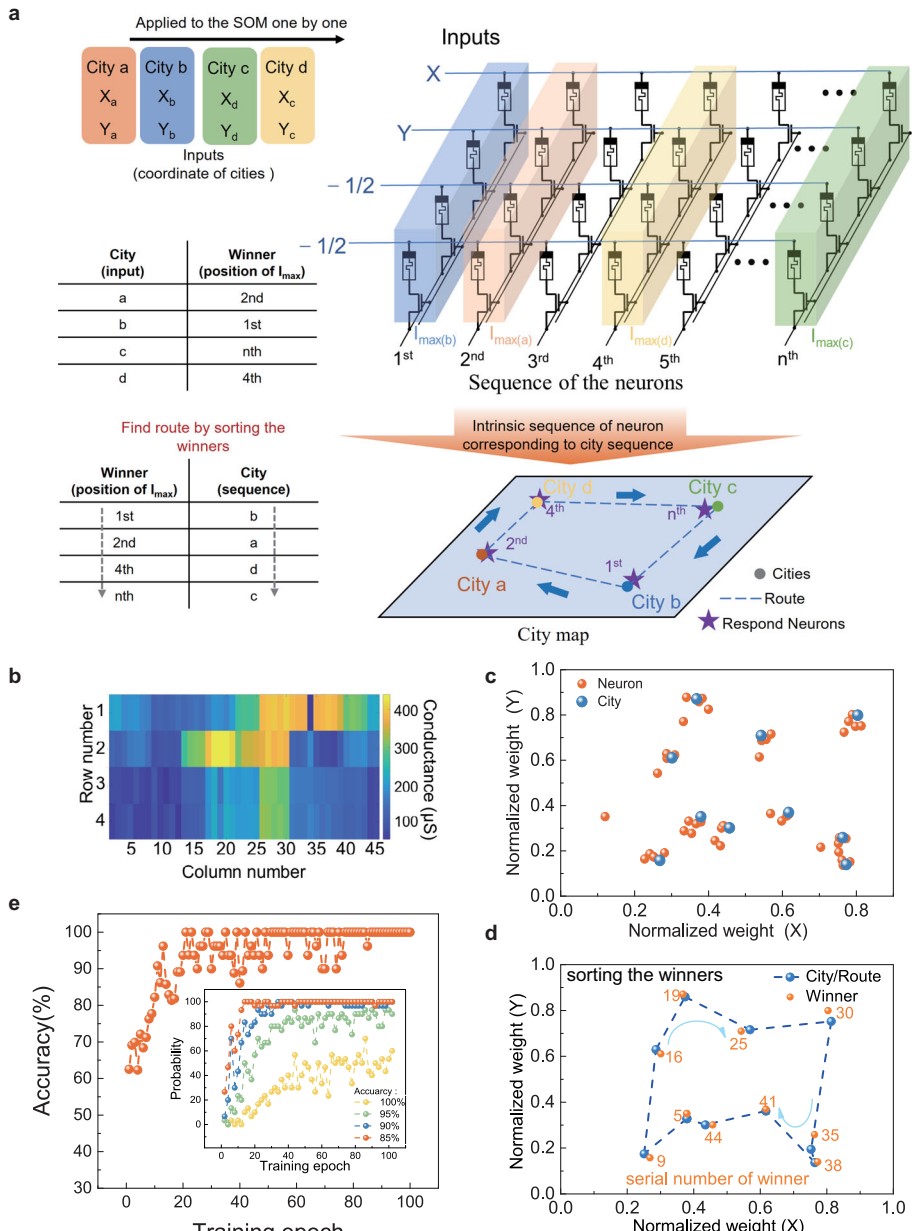

**Fig. 3 Memristor-based SOMs for TSP. a** Principle of solving TSP using memristor-based SOMs after the training process. The winner with the largest output current represents the node closest to the city. The weight updating process is similar to pushing the winner towards the city and inducing its neighborhoods to do with less intensity. This correlation between the motion of neighborhood nodes intuitively leads to a minimization of the distance, hence giving a short tour. In the testing process, all cities are applied to the SOM one by one. Normally different winners were found when applying different cities; the route can be directly found without extra configuration of the network by sorting the winners according to the intrinsic sequence of neurons in the memristor array. **b** Conductance of memristor arrays after 270 training epochs for solving TSP. **c** The city map and weight maps of all neurons after training. Blue dots represent 10 different cities, and red dots represent the normalized weights of 45 different neurons in city map. **d** Testing result of 10 cities TSP by 4 × 45 arrays. Ten different winners are found when applying to cities. And the shortest route (blue dashed lines) is determined by sorting the winners. The serial numbers of winners in the memristor array are showed by the digit next to the winner. **e** The accuracy of solving 10 city TSP by 45 nodes SOM increases with the number of training epochs. Inset shows the probabilities of different accuracies with the number of training epochs. Here the $P_{100\%}$ is the success probability that finding the shortest route.

directly correspond to the city sequence. Therefore, the route can be directly obtained without extra configuration of the network by sorting the winners according to the intrinsic sequence of neurons in the memristor array. In this example, by sorting the winner sequence in the crossbar array (the 1st column −>the 2nd column −>the 4th column− > the nth column), the obtained shortest route between chosen cities should be departed from city B via city A, city D to city C.

We experimentally solved a 10-city TSP with our memristor-based SOM, in which the coordinates of the cities were randomly generated as shown in Fig. 3b–d. The conductance map and normalized weight of the training process are shown in Fig. S9. With a gradually decreasing neighborhood function in the training process, the nodes will progressively become independent and eventually attach to different cities. After 210 training epochs, the final conductance map of the crossbar array is shown

in Fig. 3b. And Fig. 3c shows the city map and normalized weight vectors of all nodes that mapped into the city space; the blue dots represent the position of 10 cities, and the orange dots are the normalized weights of 45 different nodes. After training, the city's coordinate will be applied to the SOM one by one to find the winner neuron, just like every city will 'catch' one nearest node of the ring. Based on this method, the testing result of 10 cities is presented in Fig. 3d. The orange numbers are the serials number of the winner nodes in the whole array when 10-city coordinates are acted as inputs. The blue dash line is the shortest route calculated by our SOM, determined by the node sequence in the memristor array $(5 \rightarrow 9 \rightarrow \ldots \rightarrow 38 \rightarrow 41 \rightarrow 45)$. In the experimental training process, some neuron outputs deviate from the original positions, which is caused by the writing error and the variation of the devices. However, in our SOM, the number of nodes is three to five times the number of cities. As a result, in most cases, the extra neurons help overcome this issue.

The performance of the SOM to solve TSP is analyzed with the accuracy and probability ($P$) which are defined as Accuracy = $\frac{Shortest\_Distance}{Distance_{tested}}$ and $P_{Accuracy} = P(Distance_{tested} \leq \frac{Distance_{shortest}}{Accuracy})$, respectively. As shown in Fig. 3e, the accuracy increases with the number of training epochs and the shortest route can be obtained just after 50 training epochs. The inset of Fig. 3e shows the probabilities of different accuracies with the number of training epochs. The success probability that finding the shortest route ($P_{100\%}$) is nearly 58% after 100 training epochs. And $P_{95\%}$ can beyond 90% after 100 epochs, whereas the $P_{90\%}$ and $P_{85\%}$ can achieve nearly 100% just after 40 epochs.

The performance of the SOM with different conditions is futured studied in Fig. 4. Figure 4a shows the result of 20-node SOM with a different number of cities. The system can obtain perfect results for a six-city TSP. With the increase of the number of cities, the complexity of the problem increases, the ratio between the winner nodes and the cites and the accuracy

decrease. It is worth noting that which city should be firstly arrived is randomly selected when different cities make the same node win during the testing process. The accuracy and probability decrease due to the device writing error and the lack of nodes in the SOM. The writing-error-induced performance degradation is further confirmed in Fig. 4b. The simulation is based on a $4 \times 70$ array with different writing errors (from 0% to 5%). High accuracy and probability ($P_{95\%}$) are obtained with low writing error (below 1%). With increasing of the writing errors, the ratio between the winners and the cities decreases, which implies the deterioration of the ability to distinguish nearby cities. The accuracy decreases to 75% and $P_{95\%}$ decreases to 13% with a 5% writing error. A significant impact of writing errors on the accuracy due to the SOM does not rely on minimizing an output label error or a cost function which can provide a feedback mechanism to help network stabilization[41]. As a result, the Euclidean distance between the weight vectors and the inputs should be calculated precisely to solve the TSPs with SOM. To reduce the write errors, we adopt a write-and-verify method in our hardware SOM system. To further reduce the writing errors, multiple-device-as-one-synapse approach is used, which decreases the total writing error and increase the robustness of the network[48]. As shown in Fig. S10, we simply added more sub-arrays in the memristor array to implement multiple device synapses. For example, a three-device-weight in a 4×N weight map can be easily achieved with a 12×N array by parallelly connecting three identical sub-3×N arrays, without any extra circuit. This effective method has been tested and verified with experiment. For an 8-city TSP, based on our 20 columns(neurons) array, the experiment accuracy and $P_{95\%}$ increased from 78% to 93% and from 64% to 78%, respectively when using five devices as one weight, as shown in Fig. 4c.

Increasing the number of output nodes will also improve the performance of SOM. As presented in Fig. 4d, for a 20 cities TSP,

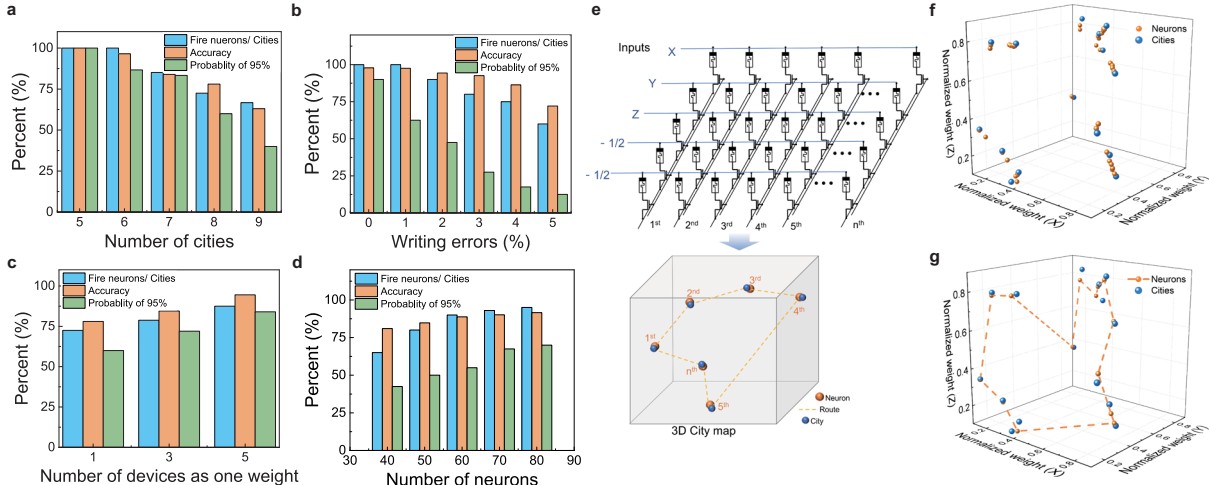

**Fig. 4 Optimization of memristor-based SOM for TSP. a** Result of the different number of cities (from 5 cities to 9 cities) TSPs with memristor-based SOM of 20 neurons. The number of cities increase, the complexity of the problem increases, and the accuracy would decrease. The SOM can perfectly solve the 5-city TSP. With the increase of the number of tested cities, the number of cities that cannot be distinguished increases. And the accuracy and the $P_{95\%}$ decreases to 68% and 36% simultaneously. **b** The impacts of the writing errors, a 20-city TSP with a $4 \times 70$ array, are simulated. The acceptable shortest route can be achieved with 0% writing error. With the increasing of wring error, the ratio between the firing neurons and the cities is decreased. And the accuracy and $P_{95\%}$ decrease to 75% and 13%with a 5% writing error. **c** Experimental result of using multiple (3 and 5) devices to represent one weight in 20 neurons SOM for 8-city TSP problems. Using more devices can decrease writing errors and increase accuracy and probability. **d** A 20-city TSP was solved in simulation to show the effectiveness of increasing the number of output nodes. With the increasing number of nodes, the ratio between firing nodes and cities is significantly increasing, whereas the accuracy and probability are simultaneously increased. **e–g** To show the potential of solving the complex problem of our system, a 15-city 3D-TSP is tested with 5 × 45 array. **e** The city map in 3D space, and the weight map of all neurons after training in city map. **f** Testing result of 15-city 3D-TSP. The blue dots represent the position of the cities. Orange dashed line is the shortest route in 3D space determined by sorting the winners when applying different cities into the SOM.

the accuracy and $P_{95\%}$ are only 74% and 39% when using 40 output nodes. Because if there are not enough nodes, some cities may be far from all the nodes or more nodes will converge to the same city, which stems from the shrinking ability of the SOM that makes nodes tend to cluster together. With the number of neurons rising from 40 to 80, the accuracy increases from 74% to 91%; whereas $P_{95\%}$ increases to 68%. It is worth noting that the accuracy and probability will not remarkably increase when continually using more neurons. As a result, to obtain a solution for N-city TSP, ~3N to 5N neurons are needed in our SOM hardware for acceptable accuracy.

Besides normal 2D-TSP, 3D TSPs can also be easily implemented by our SOM just by using three inputs in data rows (Fig. 4e) The city map in 3D space, and the weight maps of all neurons after training in of 15-city 3D-TSP solved by 5 × 45 array is presented in Fig. 4f. And Fig. 4g shows the testing result of 15-city 3D-TSP in which the blue dots represent the position of the cities. Orange dashed line is the shortest route in 3D space determined by sorting the winners. Besides, a larger-scale memristor-based SOM with ideal device performance will address TSP with a much large number of cities. Fifty-city TSP is demonstrated with simulation Fig. S11 to show the potential of solving the complex problem of our system. For hardware-implementation of large-scale SOM, one can use a single large array or multiple small arrays, and architecture-level design (which is out of the scope of this work) may be necessary for large-scale and complex problems. Our memristor-based SOM also shows the potential of energy-efficient in solving optimization problems. In solving 10-city TSP by the 12 × 45 SOM, the total energy consumption in the memristor array part is only 12.71 nJ. And the detail of energy consumption of the system has been presented in Supplementary Note 1.

In previous work, memristor-based Hopfield networks have been proposed to solve the optimization problems[49,50]. The principle of SOM and Hopfield for TSP is totally different. SOM is considered as an elastic ring. The shortest route is obtained by the shrink ability from the neighborhood function. And Hopfield minimizes the energy function. The memristors in SOM are written at each cycle which may suffer reliability issues with cycling, while in the Hopfield the meristor conducatnce are static. However, in hardware implementation solving the TSP based on Hopfield network is extremely expensive. For an N-city TSP, $N^2$ neurons and $N^4$ synapses are needed since the number of fully interconnected nodes is proportional to the square of the number of cities. And for our memristor-based SOM, even counting on the additional square rows, only around 4N neurons and 16N synapses are needed, proving that our SOM can solve more complex optimization problems with less energy consumption and hardware cost. Besides compared with our previous work about solving TSP with memristor based on Hopfield network[51], which obtained 40% successful probability with $10^4$ predetermined iteration cycles for 8-city TSP, our SOM system can achieve higher successful probability (58%) only with hundreds training epochs for 10-city TSP.

## Discussion

We have experimentally demonstrated in-situ SOM in memristor crossbar arrays. The Euclidean distance is directly calculated in the hardware by adopting additional rows of 1T1R cells. The similarities between input vectors and weight vectors are computed in one readout step without normalized weights. We have further employed the memristor-based SOM in clustering and solving the traveling salesman problems. Taking advantage of the intrinsic physical properties of memristors and the massive parallelism of the crossbar architecture, the novel memristor-based SOM hardware has advantages in computing speed, throughput, and energy efficiency, compared with current state-of-the-art

SOM implementation, as shown in Supplementary Table 2. Unlike a digital counterpart, the entire read operation is performed in a single time step, so the latency does not scale with the size of the array. The energy consumptions of the memristor device are extremely low in both inference (~40 fJ) and update (~2.42 pJ) processes. And even for compute-intensive tasks such as image segment or optimization problems (e.g., solving TSP), the memristor-based SOM can achieve a high energy efficiency system due to the small energy required in a memristor-based weight array over an all-digital system. Besides, SOM hybrid systems such as SOM-MLP[52,53], SOM-RNN[54], SOM-LSTM[55,56], show better performance and are more robust in pattern classification and prediction than simple artificial neural network systems. Our results encourage advances in the hardware implementation of unsupervised neural networks using emerging devices and provide a promising path towards machine learning or neuromorphic computing based on memristors.

## Method

**The fabrication process of 1T1R array**. The 1T1R arrays are composed of Pd/TaO$_x$/Ta memristors. The front-end and part of the back-end processes to build the transistor arrays were completed in a commercial foundry. Three different metal layers Ag/Ti/Pd (3 nm/7 nm/50 nm) were deposited by sputtering under a decent vacuum background after the argon plasma treatment to remove the native oxide layers on the via, followed by an annealing process at 300 °C for 40 min in N$_2$ atmosphere. A 2-nm Ti adhesive layer and 20 nm Pt bottom electrode was deposited by e-beam evaporator and patterned by lift-off process. A 5-nm Ta$_2$O$_5$ switching layer was deposited by sputtering followed by lift-off process for patterning. Finally, Ta/Pt (20 nm/20 nm) top electrodes was deposited and patterned by sputtering and lift-off process, respectively.

**Measurement system**. An in-house measurement system has been built to electrically read and write the 1T1R chip. The programming and computing were implemented by off-chip peripheral circuits on custom-built print circuit boards (PCBs) and MATLAB scripts, the details of the systems can be found in our early demonstrations[38–40].

## Data availability

All data needed to evaluate the conclusions in the paper are present in the paper and/or the Supplementary Materials. The source data underlying Figs. 1b, 3b–e, 4f, S3, and S9 are provided as a Source Data file. Additional data related to this paper can be requested from the authors. Source data are provided with this paper.

## Code availability

The code of the programming and computing that support the findings were implemented by MATLAB and are all based on our PCB test system. Due to the requirement of the hardware measurement system, the code is only available from the authors upon request.

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

## Acknowledgements

We thank the University of Massachusetts Amherst for the access to the memristive crossbar array and the measurement system to test the self-organizing mapping algorithms during RW's visit. This work was supported by the National Key R&D Program of China under Grant No.2018YFA0701500, the National Natural Science Foundation of China under Grant Nos. 61821091, U20A20220, 61825404, and 61732020, the Strategic Priority Research Program of the Chinese Academy of Sciences under Grant XDB44000000, and Major Scientific Research Project of Zhejiang Lab (No. 2019KC0AD02).

## Author contributions

R.W., Q.L., and T.S. designed the project. R.W., X.Z., J.W., J.L., J.Z., and Z.W. performed electrical measurements and analyzed the experimental data and results. R.W., T.S., Q.L., and M.L. wrote the manuscript. All authors discussed the results and implications and commented on the manuscript at all stages. Q.L. and M.L. supervised the research.

## Competing interests

The authors declare no competing interests.
