## [Peer Review File · Nature Communications]

REVIEWER COMMENTS

Reviewer #1 (Remarks to the Author):

The authors present an implementation of training and inference of self-organizing maps (SOM) with a memristor crossbar array. SOM is used for two applications, namely clustering and optimization. The paper is interesting and well written, the topic is relatively new. It is notable that the authors perform the training operation in-situ, which is a complicated and rare task. While the work seems technically sound, in this reviewer opinion lacks benchmark and comparison with other in-memory circuit implementation, which could significantly improve the impact of this work, especially for the final part about the travelling salesman problem (TSP). Also, the paper lacks motivation in this reviewer opinion: what are the use cases where SOM or in general unsupervised learning is preferred compared with traditional supervised learning/neural networks? Do these applications require high performance hardware?

In the following a point-to-point list of issue that should be addressed prior publication for increasing the impact:

In the SOM topography and algorithm section, it should better clarify

Why the X^2 term can be neglected and the W^2 need to be calculated even if it is a known value?

Since the squared matrix is calculated offline, how much overhead does this introduce?

Would it make sense to perform in memory only the dot product $W_data X$?

Isn't the squared term just an offset that can be add offline?

In the clustering of Iris and wine data set, how the SOM approach compare with a conventional one-layer feedforward neural network implemented in the crossbar array? Is the accuracy higher? Is the energy consumption reduced?

In the TSP optimization part, it should be clarified

Is this approach general to constraint satisfaction problems (CSP) or does it work fine only because TSP has to minimize a distance metric? Could *any* CSP be solved with SOM?

It would be interesting to show the energy of the problem evolution through the solution of the problem

How SOM behaves with local minima in the energy function?

In this reviewer opinion, is would me more appropriate to use previously used figure of merit, rather than the accuracy has defined here. A particular good metric is the probability of finding the output path, thus $P(\text{distance_test}=\text{shortest_distance})$.

Does the aforementioned probability increase with the number of cycles/iterations? In that case in would be interesting to compute the time to solution for having a 99% of success probability as function of time $TTS = t \cdot (\ln(1-0.99)) / (\ln(1-P_{\text{right}}(t)))$

How SOM compare with Hopfield neural network implemented in crosspoint array for solving TSP in terms of energy/time? In this reviewer opinion, it would be great and of high impact to encode maxCUT problem and compare the results with the table of reference [47]

Minor comments

Line 112 remove 'the' before X^2 and $W_data X$

Reference [47] should be change to Cai, F. et al. Power-efficient combinatorial optimization using intrinsic noise in memristor Hopfield neural networks. Nat Electron (2020) doi:10.1038/s41928-020-0436-6.

The authors present an implementation of training and inference of self-organizing maps (SOM) with a memristor crossbar array. SOM is used for two applications, namely clustering and optimization. The paper is interesting and well written, the topic is relatively new. It is notable that the authors perform the training operation in-situ, which is a complicated and rare task. While the work seems technically sound, in this reviewer opinion lacks benchmark and comparison with other in-memory circuit implementation, which could significantly improve the impact of this work, especially for the final part about the travelling salesman problem (TSP). Also, the paper lacks motivation in this reviewer opinion: what are the use cases where SOM or in general unsupervised learning is preferred compared with traditional supervised learning/neural networks? Do these applications require high performance hardware?

In the following a point-to-point list of issue that should be addressed prior publication for increasing the impact:

1. In the SOM topography and algorithm section, it should better clarify
 - a. Why the X^2 term can be neglected and the W^2 need to be calculated even if it is a known value?
 - b. Since the squared matrix is calculated offline, how much overhead does this introduce? Would it make sense to perform in memory only the dot product $W_{data}X$?
 - c. Isn't the squared term just an offset that can be add offline?
2. In the clustering of Iris and wine data set, how the SOM approach compare with a conventional one-layer feedforward neural network implemented in the crossbar array? Is the accuracy higher? Is the energy consumption reduced?
3. In the TSP optimization part, it should be clarified
 - a. Is this approach general to constraint satisfaction problems (CSP) or does it work fine only because TSP has to minimize a distance metric? Could *any* CSP be solved with SOM?
 - b. It would be interesting to show the energy of the problem evolution through the solution of the problem
 - c. How SOM behaves with local minima in the energy function?
 - d. In this reviewer opinion, is would me more appropriate to use previously used figure of merit, rather than the accuracy has defined here. A particular good metric is the probability of finding the output path, thus $P(\text{distance}_{test} = \text{shortest}_{distance})$.
 - e. Does the aforementioned probability increase with the number of cycles/iterations? In that case in would be interesting to compute the time to solution for having a 99% of success probability as function of time $TTS = t \cdot \frac{\ln(1-0.99)}{\ln(1-P_{right}(t))}$
 - f. How SOM compare with Hopfield neural network implemented in crosspoint array for solving TSP in terms of energy/time? In this reviewer opinion, it would be great and of high impact to encode maxCUT problem and compare the results with the table of reference [47]
4. Minor comments
 - a. Line 112 remove 'the' before X^2 and $W_{data}X$
 - b. Reference [47] should be change to Cai, F. et al. Power-efficient combinatorial optimization using intrinsic noise in memristor Hopfield neural networks. Nat Electron (2020) doi:10.1038/s41928-020-0436-6.

Reviewer #2 (Remarks to the Author):

In this work, the authors implement a memristor based self-organizing map in hardware for different applications (e.g. data clustering and solving TSP). The authors proceed further to calculate the Euclidean distance between input vectors and synapse matrix. This results are very interesting and timely, and I would recommend it for publication after addressing the following concerns.

1. The author mentioned "Compared with the previous one-row method, which use width to code the input, our multiple squire row method needs a larger area but has less time cost and less disturbance from writing errors.". Further explanation and proof needs to be furnished to justify this claim

2. In the manuscript kinds two very different kinds of application has been implemented viz., image processing and solving TSP. The message that I think the authors are trying to get across is that their system is proficient in handling varied tasks. If that is so, what is the rationale in choosing the aforementioned tasks? Apart from this, how different is the methodology for implementing these two very different tasks? Perhaps the authors should take a gander at answering this question from an algorithm or an architectural perspective.

3. What is exactly neighborhood function factor? The authors need to provide some more background regarding this.

4. Why are the authors implementing memristor based SOM for clustering? Considering that that are a lot of other clustering algorithms such as K-means why did the authors choose SOM? Are there any noticeable advantages?

5. There are some typos and grammatical mistakes that need to be dealt with.

Reviewer 1

The authors present an implementation of training and inference of self-organizing maps (SOM) with a memristor crossbar array. SOM is used for two applications, namely clustering and optimization. The paper is interesting and well written, the topic is relatively new. It is notable that the authors perform the training operation in-situ, which is a complicated and rare task. While the work seems technically sound, in this reviewer opinion lacks benchmark and comparison with other in-memory circuit implementation, which could significantly improve the impact of this work, especially for the final part about the travelling salesman problem (TSP). Also, the paper lacks motivation in this reviewer opinion: what are the use cases where SOM or in general unsupervised learning is preferred compared with traditional supervised learning/neural networks?

Answer:

Thank you for the insightful suggestions.

1) To improve the impact of our work, we have already added the comparison with other in-memory circuit implementation of SOM in our supplementary material (note 1 and table 2). Taking advantage of the intrinsic physical properties of memristors and the massive parallelism of the crossbar architecture, the novel memristor based SOM hardware has advantages in computing speed, throughput, and energy efficiency, compared with current state-of-the-art SOM implementation, as shown in Supplementary Table 2(R1).

Table R1 Comparison of the memristor based SOM and current state-of-the-art SOM hardware system.

Design	Our work	[1]	[2]	[3]	[4]
Tech.	Memristor SOM	FPGA AW-SOM	NOC- SOM	FPGA SOM	CMOS SOM
Vec. Dim.	128	3	256	3	16
Map size	8×8	5×5	16×16	16×16	16×16
Frequency (Mhz)	200	100	250	100	100
MCUPS	32768	~	18597	25344	9102
Power consumption	154.8 mW (training) 2.56 mW (testing) (for image processing: 5×64 array)	204 mW	~	~	~
in situ	Yes	~	Yes	Yes	No

The corresponding text in the supporting information has been revised as follows: For our memristor array, the maximum writing voltage is around 2.2V. 2.2V 5ns pulses could program the memristor used in this study. The frequency is around 200MHz. Due to the 128×64 size memristor array, our hardware system can implement a SOM with 8×8 map size and 128 vector dimensions. As a result, the ideal MCUPS (millions of updates per second) is around $128 \times 64 \times 200 \div 50 = 32768$.

2) And SOM normally acted as a clustering tool. By utilizing the topology and neighborhood function of the SOM, the SOM can also be solving the TSP problem. However, Hopfield solved the TSP by minimizing the energy function. So, the principles of these two methods are quite different. And we have already added the comparison between our previous work of solving TSP by Hopfield and our

memristor-based SOM, that our SOM system can achieve higher successful probability (58%) with less training epochs and less hardware cost.

3) Besides, compared with the supervised network, the unsupervised learning is more similar with human brain and has more extensive application, considering that most data and information are unlabeled in real world. Besides, unsupervised approach can cluster or pre-process the unlabeled complex data to smaller subspaces for subsequent classification through another supervised network. in the introduction. We added a few sentences to describe the advantages of unsupervised learning compared with traditional supervised learning/neural networks.

In the following a point-to-point list of issue that should be addressed prior publication for increasing the impact.

Comment 1 *In the SOM topography and algorithm section, it should better clarify a. Why the X^2 term can be neglected and the W^2 need to be calculated even if it is a known value?*

Answer: Thanks for your suggestion. The Euclidean distance can be calculated by the $Distance = x^2 + wx + w^2$. On the one hand, the X will keep constant after normalizing the input datasets in the initiation process. During the training process, in each training epoch, for all nodes the X^2 term are only depended on the input value and equal with each other as a result the X^2 term will not affect the comparison when determining the winner. However, the W will be updated in each epoch, for different

node the weight is different. As a result, the W^2 term affects the selection of the winning neuron and cannot be neglected.

The corresponding text in the manuscript has been revised as follows: **During the training process, in each training epoch, for all nodes the X^2 term are only depended on the input value and equal with each other as a result the X^2 term will not affect the comparison when determining the winner. However, the W will be updated in each epoch, for different node the weight is different. As a result, the W^2 term affects the selection of the winning neuron and cannot be neglected.**

Comment 2. *Since the squared matrix is calculated offline, how much overhead does this introduce? Would it make sense to perform in memory only the dot product $WdataX$?*

Answer: Thanks for your comments. To calculate the squared matrix and Euclidean distance, the overhead in hardware part is the extra cost in memristor array (addition rows). And in training process, the feedforward process (finding the winner) is in hardware part. And the updated weights (ΔW_{data} and $\Delta W_{squared}$) are calculated in the software. Our method does not need to normalize the weight and the only extra cost is calculating the $\Delta W_{squared}$, which introduce much less overhead compared to the previous method to calculate the Euclidean distance by normalizing the weight matrix in every training epochs Besides, only the weights of winner and the neighbors are updated and the neighborhood area gets smaller with training. Therefore, not all the squared matrixes need to be calculated, and only few nodes are updated. Based on the above analysis, the overhead of the squared matrix updates can be limited. On the other hand, the squared matrix is calculated offline, only because the limitation of our hardware measurement system that can only read and

write in one direction (write on rows and read on columns). The squired term can be directly calculated in memory based on the memristor array, as we discussed in the comment 3.

In inference process, not only the dot product $W_{data} * X$, but also the squired values can be calculated by the dot product in memory, since the inputs of the squired rows are “ $-1/2$ ”, the outputs of each column is $-1/2 * W_{squared} + W_{data} * X$. As a result, the system can directly calculate the Euclidean distance (consists of both $W_{data}X$ and $-1/2 * W_{squared}$) in memory in feedforward process of training and inference.

Comment 3. *Isn't the squired term just an offset that can be add offline?*

Answer: Thanks. Actually, the squired term can be directly calculated in on-line mode based on the memristor array⁵. As shown in Figure R1, the squired values are obtained in the memristor crossbar by sequential backward and forward read operations through the **W** matrix. Firstly, in backward reading process, a reading pulse proportional to 1 is applied to the n column. The scaled weight elements of data rows can be obtained while the squired rows are floated. Then, in forward reading process, these values are then used as inputs and fed into the data rows with the squired rows floating. The output collected at column n then corresponds to the squired value. However, because of the limitation of our hardware measurement system that can only read and write in one direction (write on rows and read on columns), the squired rows are calculated in software part in offline mode.

Figure R1 Two-steps reading method to directly calculate the squared values in memristor array

Comment 4. *In the clustering of Iris and wine data set, how the SOM approach compare with a conventional one-layer feedforward neural network implemented in the crossbar array? Is the accuracy higher? Is the energy consumption reduced?*

Answer: Thanks for your comment. SOM is a kind of widely used clustering tool. SOM is a fully unsupervised approach. 1) Compared to supervised network, our SOM can achieve competitive accuracy with lower energy consumption. Most conventional one-layer feedforward neural network in the crossbar array are based on supervised algorithm, therefore their main function is classification but not clustering. As a result, the accuracy of supervised network is normally higher, e.g., 96.08% and 94.7%. for IRIS and wine data set, respectively^{6,7}. But compared with supervised method, our SOM can achieve competitive accuracy (94.6% and 95% for IRIS and wine data set, respectively. Besides, without the back propagation process

our SOM can significantly reduce the computational complexity and energy consumption. 2) In comparison with other unsupervised cluster algorithm (e.g., K-means 93.3% for IRIS data set⁵), our system is more robust and can achieve higher accuracy(94.6%). SOM does not need any additional information at all. K-means needs to determine the number of classes beforehand. Therefore, SOM is less affected by initialization and has more generality. In training process, SOM updates the winner and the adjacent nodes, while most approaches only update a single node. Therefore, SOM is less affected by noise data. SOM also has good visualization and elegant topology diagram, which make people easier to understand and analyze.

The corresponding text in the manuscript has been revised as follows: Compared to supervised method (96.08% and 94.7% for IRIS and wine data set, respectively), our SOM can achieve competitive accuracy which is also better than memristor based K-mean system (93.3 for IRIS data set). Besides in comparison with other unsupervised cluster algorithm (e.g., K-means) SOM does not need any additional information at all. K-means needs to determine the number of classes Therefore, SOM is less affected by initialization and has more generality. In training process, SOM updates the winner and the adjacent nodes. Most approach only updates only a single node. Therefore, SOM is less affected by noise data. SOM has good visualization and elegant topology diagram, which make people easier to understand and analyze.

Comment 5. *In the TSP optimization part, it should be clarified*

*Is this approach general to constraint satisfaction problems (CSP) or does it work fine only because TSP has to minimize a distance metric? Could *any* CSP be solved with SOM?*

Answer: Thanks. Constraint satisfaction problems are the problems that a set of objects whose state must satisfy some constraints or limitations, which consist of a lot of different questions (e.g., eight queens puzzle or map coloring problem). SOM is not the general approach for all kinds of CSPs. However, not only the TSP can be solved by SOM. For example, an image segmentation problem can be casted as a CSP by interpreting the process as one of assigning labels to pixels subject to certain spatial constraints⁸ and can be solved by SOM (presented in the manuscript).

Whether a problem can be solved by SOM is limited by the topology of the SOM. Take the TSP problem as an example, due to the unique topology structure of SOM output layer (1D-ring), which is very similar with the route (closed curve) in TSP, people find that it is possible to use SOM solving TSP. In solving TSP, SOM is considered as an elastic ring. In the training process, the nodes turn to catch the cities, and due to the shrink ability from the neighborhood function, the ring trends to minimize the perimeter.

Solving other constraint satisfaction problems or combinatorial optimization problems with SOM may need utilize other similiters calculation method and design novel topology structure of the SOM, and that is still an open question.

Comment 6. *It would be interesting to show the energy of the problem evolution through the solution of the problem*

Answer: Thank you for the great suggestion. The idea of the SOM to solve the TSP is using topological structure to act as the route and utilizing the shrink ability of neighborhood function to minimize the distance. It is quite different with other methods, e.g., the Hopfield which builds the energy function and minimizes the

energy to get the best routes. As a result, the SOM do not have the energy evolution. But we added some new figures (in Comment 9) to show the probability and accuracy evolution through the solution of the problem.

Comment 7. *How SOM behaves with local minima in the energy function?*

Answer: Thanks. As we discussed in last comment. The SOM do not have energy function which is totally different with other algorithms (Hopfield). Normally the performance of the SOM to solve TSP is evaluated by the accuracy and probability, which we discuss in detail in next comment (8).

Comment 8. *In this reviewer opinion, is would me more appropriate to use previously used figure of merit, rather than the accuracy has defined here. A particular good metric is the probability of finding the output path, thus $P(\text{distance}_{\text{test}} = \text{shortestdistance})$.*

Answer: Thank you for the insightful suggestion. $\frac{\text{Shortest_Distance}}{\text{Distance}_{\text{tested}}}$ is a standard benchmark to evaluate the solving TSP ability of SOM⁹. But we also added the probability to analyze the performance of our SOM. In the revised manuscript we keep using accuracy but also added the $P_{\text{Accuracy}} = P(\text{Distance}_{\text{tested}} \leq \frac{\text{Distance}_{\text{shortest}}}{\text{Accuracy}})$, in which the probability of finding the shortest route is P_{100%}.

The corresponding text in the manuscript has been revised as follows: The performance of the SOM to solve TSP is analyzed in with the accuracy and probability (P) which are defined as accuracy = $\frac{\text{Shortest_Distance}}{\text{Distance}_{\text{tested}}}$ and $P_{\text{Accuracy}} =$

$P(\text{Distance}_{\text{tested}} \leq \frac{\text{Distance}_{\text{shortest}}}{\text{Accuracy}})$, respectively.

We also updated the figure 4(a)~(d) to show the $P_{95\%}$ with different conditions and added figure 3(e) to show the $P_{100\%}$ with the number of training epochs.

Figure 4 Optimization of memristor-based SOM for TSP. (a) Result of the different number of cities (from 5 cities to 9 cities) TSPs with memristor-based SOM of 20 neurons. The number of cities increase, the complexity of the problem increases, and the accuracy decreases. The SOM can perfectly solve the 5-city TSP. With the increase of the number of tested cities, the number of cities that cannot be distinguished increases. And the accuracy and the $P_{95\%}$ decreases to 68% and 36% simultaneously. (b) The impacts of the writing errors, a 20-city TSP with a 4×70 array, are simulated. The acceptable shortest route can be achieved with 0% writing error. With the increasing of writing error, the ratio between the firing neurons and the cities is decreased. And the accuracy and $P_{95\%}$ decrease to 75% and 13% with a 5% writing error. (c) Experimental result of using multiple (3 and 5) devices to represent one weight in 20 neurons SOM for 8-city TSP problems. Using more devices can decrease writing errors and increase accuracy and probability. (d) A 20-city TSP was solved in simulation to show the effectiveness of increasing the number of output nodes. With the increasing number of nodes, the ratio between firing nodes and cities is significantly increasing, whereas the accuracy and probability are simultaneously increased.

Comment 9. Does the aforementioned probability increase with the number of cycles/iterations? In that case it would be interesting to compute the time to solution for having a 99% of success probability as function of time $TTS = t \ln(1-0.99) \ln(1-Pr_{right}(t))$

Answer: Thanks for your great advice. We have already added the figure to show the accuracy and the probability with the increasing with the number and iterations. And we also compute the TTS of a 99% of success probability as shown in Fig.R2

The corresponding text in the manuscript has been revised as follows: As shown in Fig. 3 (e), the accuracy of increases with the number of training epochs and the shortest route can be obtained just after 50 training epochs. The inset of Fig. 3 (e) shows the probabilities of different accuracy with the number of training epochs. The success probability that finding the shortest route ($P_{100\%}$) is nearly 60% after 100 training epochs. And $P_{95\%}$ can beyond 90% after 100 epochs, whereas the $P_{90\%}$ and $P_{85\%}$ can achieve nearly 100% just after 40 epochs.

Figure 3 Memristor-based SOMs for TSP (e) The accuracy of solving 10 city TSP by 45 nodes SOM increases with the number of training epochs. Inset shows the probabilities of different accuracy with the number of training epochs. Here the $P_{100\%}$ is the success probability that finding the shortest route.

Figure R2 TTS of solving 10 city TSP by 45 nodes memristor base SOM.

Comment 10. *How SOM compare with Hopfield neural network implemented in crosspoint array for solving TSP in terms of energy/time? In this reviewer opinion, it would be great and of high impact to encode maxCUT problem and compare the results with the table of reference [47]*

Answer: Thank you for this great suggestion. Due to the unique topology structure of SOM output layer (1D-ring), which is very similar with the route (closed curve) in TSP, people find that it is possible to use SOM to solve TSP. So, SOM may not be directly used to encode maxCUT problem. And we will try to find a possible method to utilize the memristor based SOM for other applications in our future work. We have added a paragraph to compare the Hopfield and our memristor based SOM for solving TSP in the algorithm, hardware implementation and performance.

The corresponding text in the manuscript has been revised as follows: **The principle of SOM and Hopfield for TSP is totally different. SOM is considered as an elastic ring. The shortest route is obtained by the shrink ability from the neighborhood function. Hopfield minimize the energy function. In hardware implementation**

solving the TSP based on Hopfield network is extremely expensive. For an N-city TSP, N^2 neurons and N^4 synapses are needed since the number of fully interconnected nodes is proportional to the square of the number of cities. And for our memristor-based SOM, even counting on the additional squire rows, only around $4N$ neurons and $16N$ synapses are needed, proving that our SOM can solve more complex optimization problems with less energy consumption and hardware cost. Besides compared to our previous work about solving TSP with memristor based Hopfield network¹⁰, which obtained 40% successful probability with 10^4 pre-determined iteration cycles for 8-city TSP, our SOM system can achieve higher successful probability (58%) only with hundreds training epochs for 10-city TSP.

Comment 11. *Minor comments a. Line 112 remove 'the' before $X2$ and $WdataX$. Reference [47] should be change to Cai, F. et al. Power-efficient combinatorial optimization using intrinsic noise in memristor Hopfield neural networks. Nat Electron (2020) doi:10.1038/s41928-020-0436-6.*

Answer: Thanks for your advice, and we have corrected the grammar mistakes and changed the reference.

Reviewer 2

In this work, the authors implement a memristor based self-organizing map in hardware for different applications (e.g. data clustering and solving TSP). The authors proceed further to calculate the Euclidean distance between input vectors and synapse matrix. This result is very interesting and timely, and I would recommend it for publication after addressing the following concerns.

Comment 1. *The author mentioned “Compared with the previous one-row method, which use width to code the input, our multiple squire row method needs a larger area but has less time cost and less disturbance from writing errors.”. Further explanation and proof needs to be furnished to justify this claim.*

Answer: In response to your great suggestion, we have added a few sentences to explain this claim.

The corresponding text in the manuscript has been revised as follows: **In the one additional row method, which use width ($0\sim T$) to code the input ($0\sim 1$). To fit the conductance dynamic range of data row and additional row, the conductance of the device in the one additional row is W^2/L . As a result, the pulse width of the additional row should be $L*T$, which makes L times time cost than our multiple rows method. Besides, compared with the method of using one device to represent a single value, we use multiple devices to represent the W^2 , which is equivalent to average the conductance of multiple devices as one weight that can significantly reduce the impact of the writing errors of one device.**

Comment 2. *In the manuscript kinds two very different kinds of application has been implemented viz., image processing and solving TSP. The message that I think the*

authors are trying to get across is that their system is proficient in handling varied tasks. If that is so, what is the rationale in choosing the aforementioned tasks? Apart from this, how different is the methodology for implementing these two very different tasks? Perhaps the authors should take a gander at answering this question from an algorithm or an architectural perspective.

Answer: Thank you for the insightful suggestion. For these two applications, the spirit of algorithm and the architecture are quite different. We have already updated the manuscript to describe the difference.

The corresponding text in the manuscript has been revised as follows: In clustering application, people normally use the SOM with 2D planer output layer to maintain more topology information of input data. However, in solving TSP problems, the SOM with 1D-ring output layer is more common. In clustering, the SOM is a mapping tool which map the high dimension data to low dimension space, the neighborhood function is used to maintain the topology information of input data. In TSP, SOM is considered as an elastic ring, with the training process, the nodes turn to catch the cities, and due to the shrink ability from the neighborhood function, the ring trends to minimize the perimeter.

Comment 3. *What is exactly neighborhood function factor? The authors need to provide some more background regarding this.*

Answer: In the SOM the learning rate is determined by $T_i = \exp\left(-\frac{(r_c - r_i)^2}{2 \cdot \delta^2}\right)$, δ is a time-carrying parameter that guides the reduction of the neighborhood function during training. $\delta = A * \exp(-t/t_0)$, t is number of iteration and t_0 is the time constant. Here A is the neighborhood function value, which is represented to the

initial size of neighborhood area. We have defined that in the revised manuscript.

Comment 4. *Why are the authors implementing memristor based SOM for clustering? Considering that there are a lot of other clustering algorithms such as K-means why did the authors choose SOM? Are there any noticeable advantages?*

Answer: Thanks for your great advice. We have added a few sentences to show the advantages of the SOM.

The corresponding text in the manuscript has been revised as follows: Compared to supervised method (96.08% and 94.7% for IRIS and wine data set, respectively)1, 243, 44, our SOM can achieve competitive accuracy which is also better than memristor based K-mean system (93.3 for IRIS data set). Besides Compared to other unsupervised cluster algorithm (e.g., K-means) SOM does not need any additional information at all. K-means needs to determine the number of classes. Therefore, SOM is less affected by initialization and has more generality. In training process, SOM updates the winner and the adjacent nodes. Most approach only updates only a single node. Therefore, SOM is less affected by noise data. SOM has good visualization and elegant topology diagram, which make people easier to understand and analyze.

Comment 5. There are some typos and grammatical mistakes that need to be dealt with.

Answer: Thanks for your advice, and we have corrected the grammar mistakes

- 1 Cardarilli, G. C., Di Nunzio, L., Fazzolari, R., Re, M. & Spanò, S. AW-SOM, an algorithm for high-speed learning in hardware self-organizing maps. *IEEE Transactions on Circuits and Systems II: Express Briefs* **67**, 380-384 (2019).
- 2 Abadi, M., Jovanovic, S., Khalifa, K. B., Weber, S. & Bedoui, M. H. A scalable and adaptable hardware NoC-based self organizing map. *Microprocessors and Microsystems* **57**, 1-14 (2018).
- 3 Hikawa, H. & Maeda, Y. Improved learning performance of hardware self-organizing map using a novel neighborhood function. *IEEE transactions on neural networks and learning systems* **26**, 2861-2873 (2015).
- 4 Ramirez-Agundis, A., Gadea-Girones, R. & Colom-Palero, R. A hardware design of a massive-parallel, modular NN-based vector quantizer for real-time video coding. *Microprocessors and Microsystems* **32**, 33-44 (2008).
- 5 Jeong, Y., Lee, J., Moon, J., Shin, J. H. & Lu, W. D. K-means data clustering with memristor networks. *Nano letters* **18**, 4447-4453 (2018).
- 6 Hasan, R., Taha, T. M. & Yakopcic, C. On-chip training of memristor crossbar based multi-layer neural networks. *Microelectronics Journal* **66**, 31-40 (2017).
- 7 Wen, S. *et al.* Adjusting learning rate of memristor-based multilayer neural networks via fuzzy method. *IEEE Transactions on Computer-Aided Design of Integrated Circuits and Systems* **38**, 1084-1094 (2018).
- 8 Lin, W.-C., Tsao, E. C.-K. & Chen, C.-T. Constraint satisfaction neural networks for image segmentation. *Pattern Recognition* **25**, 679-693 (1992).
- 9 Angeniol, B., Vaubois, G. D. L. C. & Le Texier, J.-Y. Self-organizing feature maps and the travelling salesman problem. *Neural Networks* **1**, 289-293 (1988).
- 10 Lu, J. *et al.* Quantitatively evaluating the effect of read noise in memristive Hopfield network on solving traveling salesman problem. *IEEE Electron Device Letters* **41**, 1688-1691 (2020).

REVIEWER COMMENTS

Reviewer #1 (Remarks to the Author):

The authors did a great work revising the manuscript based on this reviewer suggestions. While the paper quality has improved, there are still some doubts concerning this reviewer, which are listed below:

1. In the previous review, this reviewer asked "In the clustering of Iris and wine data set, how the SOM approach compare with a conventional one-layer feedforward neural network implemented in the crossbar array? Is the accuracy higher? Is the energy consumption reduced?". The authors reported that:

a. SOM achieves 94.6% and 95% accuracy and supervised method achieve 96.08% and 94.7% for IRIS and wine data set, respectively. Why SOM has an irregular behaviour where a simpler problem (IRIS) has lower accuracy then a more difficult one (wine data set)? Given this irregular behavior, the initial question remains: given an amount of hardware (i.e. corsspoint arrays) why should SOM be preferred over feed forward neural networks for classification task? The suggestion is to present a case where unsupervised learning shines over supervised learning.

b. "SOM is less affected by noise data". This sentence should be proved, or a citation of previous work is required, otherwise this sentence should be removed

c. "SOM has good visualization and elegant topology diagram, which make people easier to understand and analyze." This is very important but not proved. Is SOM more 'explainable' than other classification methods? If so, a citation is required, otherwise this sentence should be removed

2. The authors added a comparison with the current state of the art in the supplementary materials. How is a large-scale problem which can't fit into a single crossbar solved? Do the authors consider a multi-array architecture for this comparison?

3. Regarding the TSP section it is now clearer to this reviewer the advantages and disadvantages of SOM versus Hopfield neural networks (HNN). A few more clarification should be added to the text

a. While it is clear that the big advantage is the better scaling of the hardware resources needed, it should be stated that the disadvantage of SOM is that memristors are written at each cycle while in the HNN the memristor conductance are static. Given that memristors suffer form reliability issues with cycling, computations where read operation are more frequent than write operation are preferred for this kind of technology

b. Given that the authors already present a work about solving TSP with HNN (reference 50), this reviewer suggest to add a direct comparison with their previous work in this paper, a figure that shows for this work and ref 50

i. Scaling of the needed resources

ii. Probability to solution

iii. Time to solution

4. Minor comment, there are still a few typos please check. For example 'accuracy' is sometimes written as 'accuarcy'.

Reviewer #2 (Remarks to the Author):

The authors have addressed my previous concerns in the revised version, which can be accepted as is now.

Reviewer 1

The authors did a great work revising the manuscript based on this reviewer suggestions. While the paper quality has improved, there are still some doubts concerning this reviewer, which are listed below:

1. In the previous review, this reviewer asked “In the clustering of Iris and wine data set, how the SOM approach compare with a conventional one-layer feedforward neural network implemented in the crossbar array? Is the accuracy higher? Is the energy consumption reduced? The authors reported that:

a. SOM achieves 94.6% and 95% accuracy and supervised method achieve 96.08% and 94.7% for IRIS and wine data set, respectively.

Why SOM has an irregular behaviour where a simpler problem (IRIS) has lower accuracy then a more difficult one (wine data set)? Given this irregular behavior, the initial question remains: given an amount of hardware (i.e. corsspoint arrays) why should SOM be preferred over feed forward neural networks for classification task? The suggestion is to present a case where unsupervised learning shines over supervised learning.

Answer: Thanks for your insightful advice. The largest difference between the SOM and supervised neural network is that SOM cannot directly classify the data. It can directly cluster the data, but classify the data by some manually definition or post-processing. For example, it only needs three output nodes using supervised feedforward neural network for iris data sets (3 classes) and each node represents one class. In our SOM we use 8×8 output, the samples in each class may fire multiple output neurons, so after clustering if we want to classify the samples, we must manually define the class to which the neurons belong. Given the same amount

of hardware, it would be preferred to use our SOM in the case that we do not have the label of the data or in the case that data set is not very complex so we only need simple post-processing. It is worth noting that for complex classify task, it also has SOM hybrid systems such as SOM-MLP, SOM-RNN, SOM-LSTM, which show better performance and are more robust in pattern classification and prediction than simple artificial neural network systems (have already introduced in discussion part of manuscript).

The accuracy of the SOM for classification task also depending on manually definition or post-processing. The difference of manually definition or post-processing can also explain why a simpler problem (IRIS) has lower accuracy than a more difficult one. We test the data clustering ability of SOM in two unsupervised situations (with/without knowing the number of classes). For IRIS, neither label nor known number of classes (fully unsupervised) were used in SOM, so we have to use the non-fire and low-fire boundary to define the class and then classify the samples. And for wine set, although we did not know the label, we already know there are three kinds of wine in the data set, so we directly define the nodes to three classes (average method), and then achieve a higher accuracy (the difference is showed in Figure R1: Supplementary Figure 3(b) and Supplementary Figure 4(a)). **We also added the description of the difference in the supporting information.** If we do not know the number of the classes in wine set, as shown in Figure R1(a), there is no clearly boundary in the output layer, which make more difficult to classify and will leads lower accuracy. Besides, IRIS data set has more overlapped sample than wine set, which will also affect the performance in SOM and lead to lower accuracy.

Figure R1 The output maps of SOM for (a) wine data set and (b) iris data set.

b. “SOM is less affected by noise data”. This sentence should be proved, or a citation of previous work is required, otherwise this sentence should be removed

Answer: Thanks for your advice, and we added a citation “Abbas OA. Comparisons between data clustering algorithms. *International Arab Journal of Information Technology (IAJIT)* 5, (2008).” in the manuscript.

c. “SOM has good visualization and elegant topology diagram, which make people easier to understand and analyze.” This is very important but not proved. Is SOM more ‘explainable’ than other classification methods? If so, a citation is required, otherwise this sentence should be removed.

Answer: Thanks for your advice, and we added a citation “Miljković D. Brief review of self-organizing maps. In: *2017 40th International Convention on Information and*

Communication Technology, Electronics and Microelectronics (MIPRO)). IEEE (2017)." in the manuscript.

2. The authors added a comparison with the current state of the art in the supplementary materials. How is a large-scale problem which can't fit into a single crossbar solved? Do the authors consider a multi-array architecture for this comparison?

Answer: Thanks for the great suggestions. We only implemented small scale problem due to the limitation of hardware measurement system with a maximum array size of 8 Kb. To solve a large-scale problem, one can use a large-scale array or use multi-array architecture. Using a large array is a directly method and may easier to achieve, however large array may cause some serious problems, e.g., large wire resistance and parasitic capacitance. And using multi-small-arrays to implement a large weight matrix is a common method in recent studies, but it needs more complex external circuit and architectural design, which are out of the scope of this manuscript. We will optimize the array structure and the measurement system to achieve multi-array architecture for larger problem in our future work.

The corresponding text in the manuscript has been revised as follows: **For hardware-implementation of large scale SOM, one can use a single large array or multiple small arrays, and architecture-level design (which is out of the scope of this work) may be necessary for large-scale and complex problems.**

3. Regarding the TSP section it is now clearer to this reviewer the advantages and disadvantages of SOM versus Hopfield neural networks (HNN). A few more clarification should be added to the text

a. While it is clear that the big advantage is the better scaling of the hardware resources needed, it should be stated that the disadvantage of SOM is that memristors are written at each cycle while in the HNN the memristor conductance are static. Given that memristors suffer form reliability issues with cycling, computations where read operation are more frequent than write operation are preferred for this kind of technology

Answer: Thanks for the great suggestions. We have added a few sentences in our manuscript. **However, the memristors in SOM are written at each cycle which may suffer reliability issues with cycling, while in the Hopfield network the memristor conductance are static.**

b. Given that the authors already present a work about solving TSP with HNN (reference 50), this reviewer suggest to add a direct comparison with their previous work in this paper, a figure that shows for this work and ref 50

i. Scaling of the needed resources

ii. Probability to solution

iii. Time to solution

Answer: Thanks for the great suggestions. We have added a figure in the supplementary material directly compare the SOM system and Hopfield system.

Supplementary Figure 12. Comparison between the memristor based Hopfield network and SOM for TSP. (a) For an N-city TSP, N^2 neurons and N^4 synapses are needed in Hopfield network. And for our memristor-based SOM, even counting on the additional squire rows, only around $4N$ neurons and $12N$ synapses are needed. (b) The time and probability to solution based on SOM for 10-city TSP and Hopfield for 8-city TSP². Our SOM system can achieve higher successful probability (58%) only with less training epochs for 10-city TSP. And the Hopfield need more than 220 cycles to become stable and can obtain 40% successful probability with 10^4 pre-determined iteration cycles for 8-city TSP.

4. Minor comment, there are still a few typos please check. For example ‘accuracy’ is sometimes written as ‘accuarcy’ .

Answer: Thanks for your advice, and we have corrected the grammar mistakes and typos, e.g., ‘accuarcy’ and ‘nuerons’ to “accuracy” and “neurons” in the manuscript.

REVIEWERS' COMMENTS

Reviewer #1 (Remarks to the Author):

The authors have done an excellent work revising the manuscript based on this reviewer suggestions.